# When Should the Treatment of Obesity in Thyroid Disease Begin?

**DOI:** 10.3390/biomedicines13010157

**Published:** 2025-01-10

**Authors:** Edyta Sutkowska, Michał Kisiel, Agnieszka Zubkiewicz-Kucharska

**Affiliations:** 1Department of Clinical Physiotherapy and Rehabilitation, Faculty of Physiotherapy, Wroclaw Medical University, 50-367 Wroclaw, Poland; 2Faculty of Medicine, Wroclaw Medical University, 50-367 Wroclaw, Poland; michal.kisiel@student.umw.edu.pl; 3Department of Pediatrics, Endocrinology, Diabetology and Metabolic Diseases, Faculty of Medicine, Wroclaw Medical University, 50-367 Wroclaw, Poland; agnieszka.zubkiewicz-kucharska@umw.edu.pl

**Keywords:** obesity, obesity treatment, thyroid, thyroid hormones

## Abstract

Obesity often coexists with thyroid diseases, and the prevalence of these disorders has been on the rise for years. While hypothyroidism can contribute to excess fat tissue, the relationship between Body Mass Index (BMI) and thyroid function hormones is bidirectional. Research confirms that fat tissue reduction can positively impact thyroid function. Thus, delaying the initiation of therapies beyond substitution treatment to achieve optimal weight reduction in individuals with thyroid dysfunction seems unwarranted. The authors summarize current knowledge on this topic in the article.

## 1. Introduction

Obesity has become a leading disorder, presenting numerous challenges for healthcare professionals and policymakers tasked with planning healthcare budgets. According to reports from the World Health Organization (WHO) [1], as of 2022, one in eight people globally were living with obesity. Tackling this pandemic is critical as obesity underpins many diseases, including those linked to shorter life expectancy, such as cardiovascular disorders [2]. While the consequences of obesity are well-documented [2,3,4], its causes remain less clear and multifaceted [4,5]. Understanding the reasons behind the dramatic rise in obesity rates [6] is essential for prevention and effective treatment. However, among individuals struggling with excess weight, myths surrounding the causes of obesity are common, often leading to either inaction or ineffective interventions. Despite significant efforts in many countries, addressing environmental factors has not yielded the anticipated slowdown in obesity rates, though some regions report a stabilization [6]. On a basic level, low physical activity and poor dietary habits are undeniably major risk factors [7,8,9]. However, obesity, as a chronic disease, arises from a complex interplay of factors, ultimately resulting in energy imbalance and fat accumulation. The perceived ineffectiveness of strict diets or increased physical activity often prompts patients to explore other potential causes for their condition, including hormonal imbalances. Conditions such as insulin resistance and hypothyroidism, which often accompany obesity, are frequently seen by patients as primary causes rather than consequences of weight gain. This perspective can influence treatment approaches and patient behavior.

Hormonal disorders, including increased insulin production due to tissue resistance to its action, can indeed significantly hinder weight loss and its maintenance [10,11]. However, it remains true that behavioral factors are the primary drivers of these disorders. Consequently, behavioral interventions continue to form the cornerstone of treatment for insulin resistance associated with obesity. Such interventions include dietary modifications, increased physical activity, and lifestyle changes, which not only address the underlying behavioral causes of obesity but also improve insulin sensitivity. This holistic approach underscores the necessity of addressing root behavioral causes to achieve sustainable health outcomes [12,13,14].

Another condition often perceived as the cause of “unjustified” (non-behavioral) weight gain is hypothyroidism. Among patients with hypothyroidism, 40–50% suffer from obesity [15], and the diagnostic process for obesity frequently begins with an evaluation of thyroid function [16]. This is justified because the thyroid gland’s role in regulating fat tissue is undisputed [17,18]. However, even minor abnormalities indicating hypothyroidism are often viewed by patients and sometimes physicians as the primary reason for excess body weight. Such an interpretation has significant implications, potentially delaying proper obesity treatment. Many individuals, even with slightly elevated Thyroid-Stimulating Hormone (TSH) levels (often without other abnormalities, known as subclinical hypothyroidism), are seen as “victims” of a slow metabolism caused by thyroid dysfunction [19,20]. This perception leads to the belief that behavioral interventions or other obesity treatments will be ineffective until thyroid function is “normalized” as reflected by in appropriate hormone levels. This misunderstanding can prevent timely and effective intervention, prolonging the challenges of managing obesity.

Meanwhile, hypothyroidism can contribute to weight gain; however, for most patients, this increase is limited to 3–5 kg [19,21]. Such an absolute value is unlikely to significantly impact a patient’s Body Mass Index (BMI). For example, a person 170 cm tall and weighing 85 kg has a BMI of 29.4, while at 80 kg, their BMI decreases to 27.7—still classified as “overweight”. This relatively minor weight change becomes more significant when higher BMIs, indicative of obesity, are involved. Confirmation of even slight thyroid dysfunction in these patients often creates a psychological barrier to initiating more assertive weight-loss measures. This hesitancy stems from the belief that thyroid dysfunction is solely responsible for excess weight and that treatment with levothyroxine will lead to weight normalization. As a result, the first approach, regardless of the severity of thyroid abnormalities, is often thyroid hormone therapy without addressing obesity effectively. This strategy can lead to frustration for both the patient and their physician as normalizing hormone levels (TSH and thyroid hormones) with substitution therapy rarely results in the anticipated weight reduction. Persistent excess fat, particularly visceral fat, even after achieving euthyroidism [22], contributes to a “chronic inflammatory state”, exposing the patient to sustained cardiovascular risks and other obesity-related diseases such as type 2 diabetes mellitus (T2DM) [23].

This raises the question of when obesity in a patient or overweight patients with abnormalities in thyroid function markers should be treated by means other than thyroid hormone substitution alone. After all, substitution therapy in itself does not constitute obesity treatment.

## 2. Relevant Section and Discussion

### 2.1. Secondary Hormonal Changes in Obesity

It is a well-established fact that an increase in TSH levels is secondary to weight gain (also changes in thyroid hormone levels) [24,25,26,27,28,29,30], and that hypothyroidism can occur as part of obesity [31]. This suggests that even a reduction in body weight, without thyroid hormone substitution, can lead to normalization of the abovementioned hormone levels [32,33,34]. Abnormalities in thyroid hormone and TSH production in obesity are most likely a reflection of disturbances in the body’s energy metabolism in the presence of excess stored energy, which leads to central resistance to thyroid hormones in these individuals [35,36,37]. Data confirm that higher TSH levels correlate with higher BMI, fasting insulin, and HOMA-IR [38]. It is also known that TSH levels are regulated by hormones and neurotransmitters involved in appetite control in the central nervous system, such as neuropeptide Y (PYY) and leptin. By stimulating the release of Thyroid-Releasing Hormone (TRH) from the hypothalamus, these compounds increase TSH secretion [39]. The insulin resistance [30] and leptin resistance [40] seen in obesity may worsen parameters that reflect thyroid function through various, often unclear, mechanisms [31].

In obesity, expanded adipose tissue becomes metabolically active, releasing pro-inflammatory cytokines, which interfere with insulin signaling pathways by promoting serine phosphorylation of insulin receptor substrate (IRS). This inhibits its activation and reduces insulin sensitivity. Elevated free fatty acids (FFAs), common in obesity, accumulate in non-adipose tissues such as the liver and muscle, leading to lipid-induced insulin resistance of the tissues. This occurs because FFAs activate several kinases, e.g., protein kinase C (PKC), which impair insulin signaling [41]. Lipid accumulation is also responsible for stress in the endoplasmic reticulum (ER), which negatively affects insulin receptor function. Finally, obesity alters the secretion of adipokines such as adiponectin and leptin. Decreased adiponectin levels reduce insulin sensitivity, while increased leptin levels, combined with leptin resistance, fail to counteract insulin resistance effectively [41]. Insulin affects thyrocyte cells by increasing the sensitivity of TSH receptors, thereby supporting the synthesis of thyroid hormones [42]. Insulin resistance can disrupt this process, leading to reduced hormone production. Chronic inflammation associated with insulin resistance also impacts thyroid hormone production. Additionally, in obesity, peripheral conversion of thyroxine (T4) to triiodothyronine (T3) is impaired, altering the T3/T4 ratio [42].

Leptin resistance makes the body less sensitive to leptin’s effects. This condition fosters excessive appetite and reduced satiety, ultimately disrupting energy balance. The mechanisms underlying leptin resistance are complex and include impaired leptin transport across the blood–brain barrier as well as dysfunction of leptin receptors and signaling pathways. The first disruption prevents leptin from effectively reaching its target site, the hypothalamus, where it regulates hunger and energy expenditure. The second one, as a consequence of a chronic inflammation in obesity, driven by excessive release of pro-inflammatory cytokines, diminishes receptor sensitivity and disrupts downstream signaling cascades, reducing leptin efficacy [43]. Leptin stimulates TRH production in the hypothalamus, which in turn affects TSH release and finally regulation of thyroid hormone secretion. However, in leptin resistance, the hypothalamus becomes less responsive to leptin signals. This reduced sensitivity disrupts the normal production of TRH and TSH, potentially leading to altered thyroid hormone levels, which can impact metabolism and energy regulation.

Additionally, increased cytokine production associated with leptin and insulin resistance heightens the risk of autoimmune thyroid diseases. This occurs through the generation of inflammatory states as observed in multiple studies [44,45]. Chronic inflammation directly impairs thyroid hormone synthesis, contributing to dysfunctions such as subclinical hypothyroidism. Leptin is also critical for balancing energy expenditure and metabolic processes, which are significantly influenced by thyroid hormones. When leptin resistance occurs, these pathways are disrupted, causing an imbalance between energy intake and expenditure. This mismatch can further impair the metabolic rate, which is closely tied to thyroid hormone activity. The interplay of leptin resistance, inflammation, and disrupted signaling underscores the complex relationship between obesity, metabolic health, and thyroid function. Understanding this interplay between insulin, leptin, obesity and thyroid hormones highlights the importance of managing both obesity and thyroid function to break the cycle of metabolic and endocrine disruptions.

Therefore, particularly in cases of mild hormonal abnormalities observed in obese individuals, their secondary nature, related to obesity, should be considered. The exception would be patients in whom there is a justified suspicion of developing hypothyroidism due to factors such as exposure to iodine therapy or following thyroidectomy.

### 2.2. Behavioral Therapies in Obesity and Hypothyroidism

Caloric reduction and increased energy expenditure are, of course, tools that enhance the likelihood of weight loss in any situation, although maintaining the effects of this seemingly simple therapy is challenging. As suggested by the authors, the causes of this difficulty should be sought in childhood among individuals suffering from obesity [46,47]. Meanwhile, the role of behaviorism, both in reducing cardiovascular risk and improving hormonal disturbances, has been proven in studies [48], which supports the validity of behavioral interventions in treating obesity, regardless of thyroid gland diseases. Incorporating behavioral therapy—such as dietary adjustments that reduce meal calorie content and promote low glycemic index foods, as well as increasing physical activity—is justified in the vast majority of obesity cases unless there are signs of severe hypothyroidism or other contraindications (e.g., uncontrolled hypertension and unstable heart disease). However, patients should be informed about the moderate to the low impact of physical activity on body weight [23]. Honesty in this matter is crucial because studies do not support the idea that the role of physical activity (when following guidelines) [49] is significant enough to substantially reduce body weight [50,51]. Typically, weight reduction, considering changes in tissue proportions (loss of fat and muscle growth), amounts to around 3–5 kg [52,53]. This is again not the change most patients expect, especially those with very high BMIs. Nevertheless, as mentioned, the role of physical activity is not just to influence body weight or proportions but also to reduce cardiovascular risk [54,55].

Recent evidence from a meta-analysis examining the impact of physical exercise in individuals with hypothyroidism confirms the safety of this form of behavioral therapy. However, it also indicates that physical activity does not have a statistically significant effect on thyroid hormone levels, such as TSH, T3, or T4 [56]. On the other hand, maintaining thyroid health might be important for encouraging regular physical activity, which, in turn, could support overall health [57]. Such findings align with other studies showing that exercise can improve cardiovascular health, muscle strength, and metabolic functions in individuals with hypothyroidism, but it does not directly correct hormonal imbalances caused by thyroid dysfunction.

Physical exercise should also be evaluated in the context of another form of behavioral therapy—diet—as it can impact appetite. Physical exercise influences appetite in a dynamic manner, depending on various factors. In the short term, physical activity can suppress hunger, particularly during high-intensity endurance exercise. This occurs through a decrease in the level of ghrelin (a hormone that stimulates appetite) and an increase in satiety hormones such as PYY and glucagon-like peptide-1 (GLP-1). However, in the long term, physical activity promotes calorie intake to meet the body’s increased energy needs. This is especially true for strength training and moderate aerobic activities [58,59].

To summarize, the studies suggest that while exercise is safe and beneficial for overall health, there are no data that supports that it may directly influence thyroid function as measured by TSH, T3, or T4 levels. Nevertheless, proper thyroid function might contribute in maintaining or increasing a person’s level of physical activity, which will ultimately be beneficial to the patient.

An obvious component of therapy is the aforementioned calorie-restricted diet. Unfortunately, as many studies show, the potential for significant weight loss, and especially maintaining that loss, using this behavioral tool is limited [60]. This is not only due to lack of willpower but also due to “reprogramming” of the body when only a limited number of calories are provided—this phenomenon is called metabolic adaptation [61,62,63]. Ultimately, patients observe diminishing effects of these behavioral interventions. Slowed metabolism, increased hunger, reduced satiety, and an increased threshold for satisfaction from eating, which are the result of genetically determined preferences for energy storage [64], favor a return to “bad” eating habits, often referred to as the “yo-yo effect”. Therefore, considering the power of both behavioral therapies and supplementation treatment for hypothyroidism in reducing fat tissue, as well as the fact that in hypothyroidism part of the excess weight is caused by water retention in the body [65], other tools should be incorporated into obesity therapy from the outset.

Nutrition should be considered not only in terms of calorie reduction, which may lead to weight loss and potential improvements in thyroid gland function. Research on the Mediterranean diet, which is one of the most well-known dietary patterns, shows that individuals following this diet have better thyroid hormone levels [66]. Studies have highlighted numerous benefits of the Mediterranean diet. Despite the reduction in BMI (if caloric restriction is accompanied), patients can expect a decreased prevalence of metabolic syndrome, T2DM, cardiovascular diseases, and certain types of cancer. Additionally, it has been associated with observed improvements in mental health.

### 2.3. Incretin Therapies and Bariatric Surgery in the Treatment of Obesity and Hypothyroidism

The discovery of incretin hormones’ role not only in regulating blood glucose levels but also in regulating body weight has been a breakthrough in the treatment of T2DM [67,68,69], as well as in managing obesity without concurrent glucose metabolism disorders [70,71].

The action of incretins affects various organs. Their peripheral effects include stimulation of the pancreas to secrete insulin (a synergistic action of GIP-glucose-dependent insulinotropic polypeptide and GLP-1-glucagon-like peptide-1) in a glucose-dependent manner. This reduces the risk of inefficient insulin secretion and protects against hyperinsulinemia as well as hypoglycemia. The hormones (and their pharmacological agonists) have opposing effects on glucagon secretion: GIP increases glucagon levels, while GLP-1 inhibits them. Regarding white adipose tissue, GIP enhances insulin sensitivity, thus not only improving glucose uptake but also improving lipid metabolism parameters by promoting lipid buffering and promoting lipid storage. This prevents ectopic fat deposition. In the gastrointestinal tract, GLP-1 primarily slows upper gastrointestinal motility, discouraging the consumption of large food quantities, and contributing to the effectiveness of weight-loss therapies. In the central nervous system, both hormones reduce food intake, with GLP-1 additionally enhancing satiety, and promoting the selection of less caloric foods [67,68,69,70,71]. This multi-organ action addresses the complex nature of obesity, aiding not only in weight loss but also in maintaining the achieved results.

Initially, there were concerns about using incretin hormone agonists (glucagon-like peptide-1 or dual-acting glucose-dependent insulinotropic polypeptide and glucagon-like peptide-1) in patients with thyroid diseases. However, clinical studies have dispelled these concerns as they have not confirmed a significant risk of medullary thyroid cancer (MTC) in this patient group [72]. While use of these therapies may be problematic in individuals with a family history of MTC, other thyroid conditions should not be considered a contraindication for their use. The prolonged oncogenesis process (except in cases of hereditary MTC) suggests that people undergoing these therapies should be considered to be monitored regularly [73]; however, monitoring for the occurrence of MTC in patients taking GLP-1RAs (glucagon-like peptide-1 receptor agonists) is not recommended by the Food and Drug Administration (FDA). This monitoring is justified as the relatively short duration of incretin use in medicine may not yet fully reveal whether a cancer risk actually exists. Use of GLP-1RAs in patients with a personal or family history of MTC or multiple endocrine neoplasia (MEN) type 2 is not recommended. Meanwhile, the effectiveness of these drugs in promoting fat loss is substantial. There is also solid evidence supporting the use of GLP-1RAs for weight management, even without comprehensive lifestyle interventions, e.g., when applied for adults with pre-existing cardiovascular disease, which of course should not exempt physicians from promoting healthy behavior. The introduction of tirzepatide, in particular, has meant that some patients who might otherwise need bariatric surgery can benefit from “last-chance pharmacological treatment” before considering surgical options. Weight loss with the most commonly used incretin hormone agonist therapies ranges from 5% to >20% of baseline body weight. Such reductions offer significant benefits, modifying the risk of cardiovascular diseases [74,75] and T2DM. In patients already diagnosed with T2DM, this treatment significantly increases the likelihood of achieving disease remission [76]. While the Diabetes Remission Clinical Trial (DIRECT) study [76] showed that diabetes remission could be achieved through behavioral weight loss, in everyday clinical practice, the ability to monitor patients and provide individualized behavioral treatment that is effective long term is lacking.

This also led to the introduction of surgical procedures for the treatment of obesity [77]. This form of therapy remains the most effective, and depending on the type of procedure, patients lose between 15 and 80% of their excess body weight (EBW). The benefits include not only a reduction in fat tissue but also an improvement in metabolism, likely due to enhanced incretin hormone secretion [78]. Unfortunately, hypothyroidism is one of the contraindications for bariatric procedures, although the severity of thyroid dysfunction should be considered when deciding on surgical intervention. Subclinical hypothyroidism is not an absolute contraindication to bariatric surgery. A thorough preoperative evaluation and appropriate management are crucial for ensuring safety and optimizing outcomes. It is always advisable to collaborate closely with an endocrinologist to tailor the approach to the patient’s specific needs [69,70].

Thus, the intensity of treatment and the type of proposed interventions should correspond to the degree of advancement of the obesity disease in the patient. Regardless of the method used to achieve weight loss, it can also lead to the normalization of thyroid function parameters as confirmed by studies [33,34,79,80,81].

It is important to realize that fat loss is often accompanied by muscle mass loss. Therefore, regardless of whether pharmacotherapy or surgical treatment is implemented, it is crucial to ensure the patient maintains an appropriate level of physical activity and a balanced diet. This approach is an integral part of comprehensive obesity management, irrespective of coexisting thyroid dysfunctions.

### 2.4. Other Obesity Treatment

In the treatment of obesity, medications that inhibit digestive enzymes responsible for breaking down fats in the gastrointestinal tract can also be used, reducing their absorption. An example of such a medication is orlistat, which can lead to a reduction of 5–10% of initial body weight. Another option is a combination drug consisting of naltrexone and bupropion (Mysimba), which acts on the central nervous system, reducing appetite and controlling food cravings. The total weight reduction achieved with this medication is similar, amounting to approximately 10% of initial body weight [82]. The effects of both these medications may be less pronounced compared to newer drugs and procedures used in the treatment of obesity as discussed above. Additionally, the confirmed cardiovascular benefits associated with incretin-based therapies and bariatric surgery make orlistat and Mysimba second-line treatment options. Although, the latter can be considered as the first therapy for obese patients with a predominance of so-called emotional hunger and/or nicotine addiction [82].

### 2.5. Hyperthyroidism and Being Overweight

Inadequate weight gain is mainly associated with hypothyroidism. However, it should be noted that various diseases of the thyroid gland can influence the rate of fat deposition and, in combination with other disorders, contribute to the accumulation of excess body fat. In hyperthyroidism, weight loss is usually observed. The excess thyroid hormones accelerate metabolism, increasing the rate of calorie burning and energy expenditure by the body. However, in some cases, individuals with hyperthyroidism may experience weight gain, which is less common but possible. The reasons for this include situations where, despite the excess of thyroid hormones, metabolic changes may occur that promote fat storage, such as the aforementioned reduced tissue sensitivity to insulin (e.g., due to limited physical activity). Additionally, stress associated with hyperthyroidism and changes in eating habits can, despite the overactive thyroid, promote weight gain (hunger episodes leading to the consumption of more calories). Treatment of hyperthyroidism, both pharmacological and surgical, can lead to temporary dysregulation of thyroid hormone levels. After their normalization, a slowing of metabolism is observed (compared to the state of hyperthyroidism), which, if accompanied by persistence of poor behavioral habits, can lead to weight gain. Moreover, the process of adapting to a new metabolic state after treatment can be challenging, and this transition phase may result in fluctuations in body weight [83].

## 3. Conclusions

Given frequent issues with obesity in patients with hypothyroidism, early intervention with a comprehensive approach to obesity management seems reasonable. The table (Table 1) summarizes the most important changes that accompany obesity and their improvement depending on the therapy used as was presented in detail in the previous sections of this paper. Comprehensive intervention is further justified by the proven cardiovascular benefits, which include improvements in lipid metabolism and blood pressure—common concerns in patients with thyroid disorders. The choice of therapy and the intensity of intervention should, therefore, depend not only on the degree of obesity but also on comorbid disorders, including severe vascular complications. Previous attempts by the patient to reduce body weight should also be considered, along with an assessment of how much progress they have made toward achieving their goals.

Young individuals without complications of obesity-related disease should always begin therapy with behavioral modification. However, they should also be informed that, in case of unsatisfactory results (which is more likely with a high BMI), there are many additional options available to them.

Older individuals with multiple comorbidities should also be encouraged to engage in behavioral interventions, though the likelihood that these efforts alone will sufficiently address weight issues is lower. Therefore, in this group, earlier consideration of pharmacotherapy or surgery may be warranted, provided there are no contraindications.

Substitution therapy for hypothyroidism should be carried out concurrently, and patients should be informed that it is not the primary treatment for obesity.

Also, in the case of treating hyperthyroidism, it is important to be aware of the potential risk of uncontrolled weight gain. For this reason, behavioral interventions should be implemented to eliminate any harmful habits that the patient may have developed during untreated hyperthyroidism.

When considering at what stage of hypothyroidism treatment it is appropriate to introduce significant fat reduction strategies, one must take into account the effectiveness of thyroid hormone substitution leading to euthyroidism and the power of behavioral therapy. These approaches should then be compared with the expectations of both the physician and the patient. Together, these two interventions show moderate effectiveness in weight loss, which should prompt active pharmacological treatment or bariatric surgery in most patients. These therapies should complement hormone substitution and behavioral therapy as soon as possible. In this context, a multi-faceted approach—integrating pharmacological, behavioral, and possibly surgical interventions—will likely lead to the best long-term results for managing both obesity and hypothyroidism, improving overall health outcomes for these patients.

### Future Directions

Given the lack of studies on the use of therapies commonly applied to reduce fat mass in individuals with untreated hypothyroidism, it seems both reasonable and safe to apply these treatments primarily in cases of subclinical hypothyroidism. A promising challenge would be to conduct randomized clinical trials for individuals with thyroid disorders and obesity, particularly when the patient is not in a state of euthyroidism. This is because previous research has typically excluded participants whose thyroid condition was uncontrolled or symptomatic.

Conducting such trials would help clarify the effects of standard obesity treatments (e.g., behavioral therapies and pharmacological agents) in patients with active thyroid dysfunction, and potentially offer new insights into optimizing treatment strategies for this specific population.

## Figures and Tables

**Table 1 biomedicines-13-00157-t001:** Obesity-related changes and the impact of various factors/therapies.

Disorders Associated with Obesity	Impact of Thyroid Hormone Replacement Therapy	Impact of Physical Activity	Impact of Diet	Impact of Incretin Hormone Analogues	Impact of Bariatric Surgery
↑ appetite	↔ or ↗ *at the beginning of the treatment*	↙ or ↔ or ↗ ↙ aerobic ↗ resistance ↗ intensive resistance ↙ intensive endurance	↑ carbohydrates ↔ fats ↔ proteins ↑ caloric restriction	↓	↓
↓ satiety	↔ or ↗	↑ *mainly aerobic exercises*	↓ carbohydrates ↔ or ↗ fats ↑ proteins ↑ fiber	↑	↑
↓ energy expenditure	↑	↑	↔ fats, carbohydrates ↗ proteins, fiber ↓ caloric restriction	↗	↓
↑ ectopic fat accumulation	↔ or ↙	↓	↓ *caloric restriction*	↓	↓
↑ inflammation	↙ *Related to hypothyroidism and body mass correction*	↓	↓	↓	↓
↓ pancreatic function	↔ or ↗	↑	↑	↑	↑
↑ blood pressure	↓	↓	↓ or ↔	↓	↓
↑ blood lipids	↓	↓	↓	↓	↓
↑ Thyroid-Stimulating Hormone	↓	↔	↓ or ↔ *caloric restricion*	↓ or ↔ or ↗ ?	↙

↓ decreased; ↑ increased; ↙ mildly decreased; ↗ mildly increased; ↔ no impact; ? unknown.

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
