# Peer review of "When Should the Treatment of Obesity in Thyroid Disease Begin?"

_biomedicines, 2025, doi:10.3390/biomedicines13010157_

Round 1
Reviewer 1 Report
Comments and Suggestions for Authors
Comments and suggestions for Authors:
Thank you for the opportunity to review this article.
The Authors have explored the treatment of obesity in thyroid disease which is a very interesting issue. They drew attention to errors that may be made in the treatment of obese people with hypothyroidism.
The authors demonstrated the importance of multi-aspect integration of pharmacological, behavioral, and possibly surgical interventions to lead to the best long-term results in the treatment of both obesity and hypothyroidism.
The publication is of high substantive value and offers an in-depth treatment of the subject. The conclusions drawn are intriguing, and the entire work can serve as a starting point for further research by other scholars in this area.
However, greater attention to detail is needed for increased accuracy.
Minor errors:
Line 26 – please add the space
Line 62 – please correct: „inappropriate”
Line 90 – please correct: „to the normalization”
Line 90 – please change the phrase: „mention above”
Line 105 – please correct: „such as the liver”
Line 108 – please correct: „which negatively affects”
Line 108 – add the comma after „finally”
Line 123 – please correct: „consequence of a chronic inflammation”
Line 151 – please correct: „by the authors”
Line 158 – delete the comma
Line 160 – please correct: „to the low”
Line 187 – please correct: „which supports”
Line 189 – please correct: „give a lot”
Line 188-189 – please change this sentence
Line 220 – please correct: „hypoglycemia”
Line 226 – please correct: „and contributing”
Line 228 – please correct: „promoting”
Line 243 –„personal of family” or “personal or family” ?
Line 264 – please correct „procedures”
Author Response
Dear Editor
Dear Reviewers
We sincerely thank you for the thorough evaluation of our manuscript and your detailed comments. All your remarks are invaluable to us and help us strive for even better preparation of the topic we have undertaken. All the changes suggested by the Reviewers have been highlighted in red in the text, and below we briefly address them. After reading your reviews, we concluded that it would be worthwhile to add one more short chapter dedicated to other oral medications, which are currently less commonly used, but may have a place in the treatment of obesity coexisting with thyroid diseases. Therefore, Chapter 2.4 ( “Other obesity treatment “) has been added, along with the relevant literature.
The answer to the Reviewers.
Rev no 1
Comments and Suggestions for Authors
Thank you for the opportunity to review this article.
The Authors have explored the treatment of obesity in thyroid disease which is a very interesting issue. They drew attention to errors that may be made in the treatment of obese people with hypothyroidism.
The authors demonstrated the importance of multi-aspect integration of pharmacological, behavioral, and possibly surgical interventions to lead to the best long-term results in the treatment of both obesity and hypothyroidism.
The publication is of high substantive value and offers an in-depth treatment of the subject. The conclusions drawn are intriguing, and the entire work can serve as a starting point for further research by other scholars in this area.
However, greater attention to detail is needed for increased accuracy.
Minor errors:
Line 26 – please add the space
Line 62 – please correct: „inappropriate”
Line 90 – please correct: „to the normalization”
Line 90 – please change the phrase: „mention above”
Line 105 – please correct: „such as the liver”
Line 108 – please correct: „which negatively affects”
Line 108 – add the comma after „finally”
Line 123 – please correct: „consequence of a chronic inflammation”
Line 151 – please correct: „by the authors”
Line 158 – delete the comma
Line 160 – please correct: „to the low”
Line 187 – please correct: „which supports”
Line 189 – please correct: „give a lot”
Line 188-189 – please change this sentence
Line 220 – please correct: „hypoglycemia”
Line 226 – please correct: „and contributing”
Line 228 – please correct: „promoting”
Line 243 –„personal of family” or “personal or family” ?
Line 264 – please correct „procedures”
Answer- All corrections have been made and highlighted in the text. We sincerely thank you for pointing out the errors and mistakes with such precision.
Rev no 2
Comments and Suggestions for Authors
This study represents a cross-section of knowledge related to obesity and hypothyroidism well explaining this two-way relationship. The authors also provided a nice overview with other hormonal conditions, as well as existing therapies in the treatment of this widespread pathology.
The paper is well written but could be improved structurally.
In my opinion, the author should explain in more detail the information given in the Table 1. as it is a part of the Conclusion section and somehow represents the basis for the decision tree and what is it that they recommend in conclusion. The conclusion should be stronger and more specific.
Answer- Thank you very much for this remark. We have expanded the Conclusions section and provided a more detailed interpretation of the table as well as the main text above.
Minor corrections:
Page: 2, 3; lanes: 97, 98. It seem to be a mistake while explaining TRH which is thyroid releasing hormone, not thyroid stimulating hormone
Answer- this has been corrected
Page 5, lane: 220; typo: hypoglycaemia
Answer- this has been corrected
In addition to the reviewers' comments, the following information has been added in accordance with the editor's suggestion:
- Page 3, line 105. Authors mentioned that “FFAs activate stress kinase …” It is not clear what kinases are these. The corresponding references are not presented.
Answer: line 106: This occurs because FFAs activate several kinases, e.g. protein kinase C (PKC), which impair insulin signaling [42].
- Page 3, line 111: “Insulin affects thyroid cells by increasing the sensitivity of TSH receptors…”
It seems that authors meant “thyroid cells” to “thyroid follicular cells”. It needs to be clarified
as there are several cell types in the thyroid gland. The reference of 42 is not supporting the
statement. Please add a reference that supports the statement.
We are very sorry for the mistake we change the number of the references for the correct one.
Answer: line 111: Insulin affects thyrocytes cells by increasing the sensitivity of TSH receptors, thereby sup-porting the synthesis of thyroid hormones [43].
- Page 5, line 240 and 245: GLP-1RA was not spelled out in line 240 but was in line 245 .
Answer: We apologize for this oversight - it has been corrected.
- The authors in the review article entitled “When should the treatment of obesity in thyroid disease begin?” summarized current knowledge on the complication of obesity and thyroid disorders. The review article updated current knowledge and summarized well from the view of obesity and thyroid disorders. The article will be interesting to the audience in the field. Thyroid disorders include not only hypothyroidism but also hyperthyroidism. Authors described thyroid disorder as hypothyroidism mainly in the article, but not mentioned hyperthyroidism. It would be good to mention the reason you did not discuss about hyperthyroidism in the article, otherwise title mislead the leaders. There is a strong association of obesity with chronic low-grade inflammation, it will be worthwhile to describe the association with hypo- or hyperthyroidism.
Answer: Line 281: The sub-section “2.5. Hyperthyroidism and overweight” has been added
- Table 1, which is a part of major conclusion of the manuscript, should be explained with more information
Answer: Line 307: The extra information has been added.
We hope that the current version of the article meets the criteria necessary for publication.
With kind regards
Team leader and corresponding author.

Reviewer 2 Report
Comments and Suggestions for Authors
This study represents a cross-section of knowledge related to obesity and hypothyroidism well explaining this two-way relationship. The authors also provided a nice overview with other hormonal conditions, as well as existing therapies in the treatment of this widespread pathology.
The paper is well written but could be improved structurally.
In my opinion, the author should explain in more detail the information given in the Table 1. as it is a part of the Conclusion section and somehow represents the basis for the decision tree and what is it that they recommend in conclusion. The conclusion should be stronger and more specific.
Minor corrections:
1. Page: 2, 3; lanes: 97, 98. It seem to be a mistake while explaining TRH which is thyroid releasing hormone, not thyroid stimulating hormone
Page 5, lane: 220; typo: hypoglycaemia
Author Response

(The authors gave the same response as above.)

Round 2
Reviewer 2 Report
Comments and Suggestions for Authors
I think that the corrections made by the author are significantly improved and I suggest the revised version are ready for the publication now.